# Occurrence and transmission potential of asymptomatic and presymptomatic SARS-CoV-2 infections: Update of a living systematic review and meta-analysis

Diana Buitrago-Garcia[1,2], Aziz Mert Ipekci[1], Leonie Heron[1], Hira Imeri[1], Lucia Araujo-Chaveron[3,4], Ingrid Arevalo-Rodriguez[5], Agustín Ciapponi[6], Muge Cevik[7], Anthony Hauser[1], Muhammad Irfanul Alam[3], Kaspar Meili[8], Eric A. Meyerowitz[9], Nirmala Prajapati[10], Xueting Qiu[11], Aaron Richterman[12], William Gildardo Robles-Rodriguez[13], Shabnam Thapa[14], Ivan Zhelyazkov[15], Georgia Salanti[1], Nicola Low[1]*

1 Institute of Social and Preventive Medicine, University of Bern, Bern, Switzerland, 2 Graduate School of Health Sciences, University of Bern, Bern, Switzerland, 3 EHESP French School of Public Health, Paris and Rennes, France, 4 Institut Pasteur, Paris, France, 5 Clinical Biostatistics Unit, Hospital Universitario Ramon y Cajal, IRYCIS, CIBER of Epidemiology and Public Health, Madrid, Spain, 6 Instituto de Efectividad Clínica y Sanitaria (IECS-CONICET), Buenos Aires, Argentina, 7 Division of Infection and Global Health Research, School of Medicine, University of St. Andrews, Fife, Scotland, United Kingdom, 8 Department of Epidemiology and Global Health, Umeå University, Umeå, Sweden, 9 Division of Infectious Diseases, Montefiore Medical Center, Bronx, New York, New York, United States of America, 10 Université Paris-Saclay, Paris, France, 11 Center for Communicable Disease Dynamics, Department of Epidemiology, Harvard TH Chan School of Public Health, Boston, Massachusetts, United States of America, 12 Division of Infectious Diseases, University of Pennsylvania, Philadelphia, Pennsylvania, United States of America, 13 Fundación Universitaria de Ciencias de la Salud, Bogotá, Colombia, 14 Manchester Centre for Health Economics, University of Manchester, Manchester, United Kingdom, 15 University of Sheffield, Sheffield, United Kingdom

☯ These authors contributed equally to this work.

* nicola.low@ispm.unibe.ch

**Data Availability Statement:** All data and code used to display and synthesise the results are

## Abstract

### Background

Debate about the level of asymptomatic Severe Acute Respiratory Syndrome Coronavirus 2 (SARS-CoV-2) infection continues. The amount of evidence is increasing and study designs have changed over time. We updated a living systematic review to address 3 questions: (1) Among people who become infected with SARS-CoV-2, what proportion does not experience symptoms at all during their infection? (2) What is the infectiousness of asymptomatic and presymptomatic, compared with symptomatic, SARS-CoV-2 infection? (3) What proportion of SARS-CoV-2 transmission in a population is accounted for by people who are asymptomatic or presymptomatic?

### Methods and findings

The protocol was first published on 1 April 2020 and last updated on 18 June 2021. We searched PubMed, Embase, bioRxiv, and medRxiv, aggregated in a database of SARS-CoV-2 literature, most recently on 6 July 2021. Studies of people with PCR-diagnosed

available at https://github.com/leonieheron/LSR_Asymp_v5.

**Funding:** This study was funded by the Swiss National Science Foundation http://www.snf.ch/en (NL: 320030_176233); the European Union Horizon 2020 research and innovation programme https://ec.europa.eu/programmes/horizon2020/en (NL: 101003688); the Swiss government excellence scholarship https://www.sbfi.admin.ch/sbfi/en/home/education/scholarships-and-grants/swiss-government-excellence-scholarships.html (DBG: 2019.0774) and the Swiss School of Public Health Global P3HS stipend https://ssphplus.ch/en/ (DBG). The funders had no role in study design, data collection and analysis, decision to publish, or preparation of the manuscript.

**Competing interests:** I have read the journal's policy and the authors of this manuscript have the following competing interests: NL is an academic editor at PLOS Medicine, received the grant funding from the organisations declared above, is a member of the Swiss National COVID-19 Science Task Force, and is on the scientific board of Sefunda, a start-up company that develops point-of-care diagnostics for sexually transmitted infections. DBG receives funding from the organisations declared above. AMI, LH, HI receive salary support from the grants to NL from the Swiss National Science Foundation 320030_176233, and/or European Union H2020 101003688

**Abbreviations:** CI, confidence interval; COVID-19, Coronavirus Disease 2019; IQR, interquartile range; RT-PCR, reverse transcription PCR; SARS-CoV-2, Severe Acute Respiratory Syndrome Coronavirus 2.

SARS-CoV-2, which documented symptom status at the beginning and end of follow-up, or mathematical modelling studies were included. Studies restricted to people already diagnosed, of single individuals or families, or without sufficient follow-up were excluded. One reviewer extracted data and a second verified the extraction, with disagreement resolved by discussion or a third reviewer. Risk of bias in empirical studies was assessed with a bespoke checklist and modelling studies with a published checklist. All data syntheses were done using random effects models. Review question (1): We included 130 studies. Heterogeneity was high so we did not estimate a mean proportion of asymptomatic infections overall (interquartile range (IQR) 14% to 50%, prediction interval 2% to 90%), or in 84 studies based on screening of defined populations (IQR 20% to 65%, prediction interval 4% to 94%). In 46 studies based on contact or outbreak investigations, the summary proportion asymptomatic was 19% (95% confidence interval (CI) 15% to 25%, prediction interval 2% to 70%). (2) The secondary attack rate in contacts of people with asymptomatic infection compared with symptomatic infection was 0.32 (95% CI 0.16 to 0.64, prediction interval 0.11 to 0.95, 8 studies). (3) In 13 modelling studies fit to data, the proportion of all SARS-CoV-2 transmission from presymptomatic individuals was higher than from asymptomatic individuals. Limitations of the evidence include high heterogeneity and high risks of selection and information bias in studies that were not designed to measure persistently asymptomatic infection, and limited information about variants of concern or in people who have been vaccinated.

## Conclusions

Based on studies published up to July 2021, most SARS-CoV-2 infections were not persistently asymptomatic, and asymptomatic infections were less infectious than symptomatic infections. Summary estimates from meta-analysis may be misleading when variability between studies is extreme and prediction intervals should be presented. Future studies should determine the asymptomatic proportion of SARS-CoV-2 infections caused by variants of concern and in people with immunity following vaccination or previous infection. Without prospective longitudinal studies with methods that minimise selection and measurement biases, further updates with the study types included in this living systematic review are unlikely to be able to provide a reliable summary estimate of the proportion of asymptomatic infections caused by SARS-CoV-2.

## Review protocol

Open Science Framework (https://osf.io/9ewys/)

## Author summary

### Why was this study done?

- The proportion of people who will remain asymptomatic throughout the course of infection with Severe Acute Respiratory Syndrome Coronavirus 2 (SARS-CoV-2), the cause of Coronavirus Disease 2019 (COVID-19), is debated.

- Studies that assess people at just 1 time point overestimate the proportion of true asymptomatic infection because those who go on to develop COVID-19 symptoms will be wrongly classified as asymptomatic, but other types of study might underestimate the proportion if, for example, people with symptoms are more likely to be included in a study population.

- The number of published studies about SARS-CoV-2 is increasing continuously, types of studies are changing and, since 2021, vaccines have become available, and variants of concern have emerged.

## What did the researchers do and find?

- We updated a living systematic review through 6 July 2021, using automated workflows that speed up the review processes, and allow the review to be updated when relevant new evidence becomes available.

- In 130 studies, we found an interquartile range of 14% to 50% (prediction interval 2% to 90%) of people with SARS-CoV-2 infection that was persistently asymptomatic; owing to heterogeneity, we did not estimate a summary proportion.

- Contacts of people with asymptomatic SARS-CoV-2 infection are less likely to become infected than contacts of people with symptomatic infection (risk ratio 0.32, 95% CI 0.16 to 0.64, prediction interval 0.11 to 0.95, 8 studies).

## What do these findings mean?

- In studies published up to mid-2021, most people with SARS-CoV-2 were not persistently asymptomatic, and asymptomatic infection was less infectious than symptomatic infection.

- In the presence of high between-study variability, summary estimates from meta-analysis may be misleading and prediction intervals should be presented.

- Future studies about asymptomatic SARS-CoV-2 infections caused by variants of concern and in people with immunity following vaccination or previous infection should be specifically designed, using methods to minimise biases in the selection of study participants and in ascertainment, classification, and follow-up of symptom status.

## Introduction

There is ongoing debate about the true proportion of Severe Acute Respiratory Syndrome Coronavirus 2 (SARS-CoV-2) infection that remains asymptomatic [1]. A well-recognised source of overestimation arises when people without symptoms at the time of testing are reported as having asymptomatic infection, with such cross-sectional studies often reporting percentages of 80% or more [2,3]. These studies overestimate the proportion of persistently asymptomatic infection because they misclassify people with so-called presymptomatic infection, who will develop symptoms of Coronavirus Disease 2019 (COVID-19) if reassessed after

an adequate follow-up period [1]. Other sources of bias can result in over- or underestimation of the proportion with persistent asymptomatic infections, even when participants are adequately followed up [1]. For example, studies that assess a limited range of symptoms could overestimate the proportion asymptomatic through misclassification if they do not ask participants about all possible symptoms. Since COVID-19 was first identified as a viral pneumonia, the spectrum of symptoms has grown to include gastrointestinal symptoms and disturbances of smell and taste [1]. On the other hand, selection bias would be expected to underestimate the proportion with asymptomatic SARS-CoV-2 if people with symptoms are more likely to be tested for SARS-CoV-2 infection than those without symptoms [4].

Accurate estimates of the proportions of true asymptomatic and presymptomatic infections are needed to determine the balance and range of control measures [5]. Recognition of asymptomatic and presymptomatic infections showed the importance of control measures such as physical distancing, active case-finding through testing of asymptomatic people [6], and the need for rapid quarantine [7] in the first waves. Since late 2020, vaccines have become available [8] and several SARS-CoV-2 variants of concern have spread internationally, with varying viral characteristics [9]. The number of published studies about SARS-CoV-2 is increasing continuously, and the types of published studies are also changing [10], including the designs of studies about asymptomatic infection. In systematic reviews of studies published to April 2021, reported point estimates from random effects meta-analysis models range from 17% to 41% [11–16]. Authors of these reviews typically report values of the $I^2$ statistic >90% [17,18], but heterogeneity is often not explored in detail and prediction intervals, which give information about sampling error and variability between studies, are recommended but rarely reported [17,19,20]. In this fifth version of our living systematic review [21], we aimed to improve and understand the changing evidence over time for 3 review questions: (1) Among people who become infected with SARS-CoV-2, what proportion does not experience symptoms at all during their infection? (2) What is the infectiousness of people with asymptomatic and presymptomatic, compared with symptomatic SARS-CoV-2 infection? (3) What proportion of SARS-CoV-2 transmission is accounted for by people who are either asymptomatic throughout infection or presymptomatic?

## Methods

We conducted an update of a living systematic review, a systematic review that provides an online summary of findings and is updated when relevant new evidence becomes available [22]. The protocol, which describes modifications for each version, was first published on 1 April 2020 and amended for this fifth version on 18 June 2021 (https://osf.io/9ewys/). Previous versions have been posted as preprints [21,23] and published as a peer-reviewed article [12]. We report our findings according to statements on preferred reporting items for systematic reviews and meta-analyses 2020 (S1 PRISMA Checklist) [24] and on synthesis without meta-analysis in systematic reviews (SWiM) [25]. Ethics committee review was not required for this review. Box 1 shows our definitions of symptoms, asymptomatic infection, and presymptomatic status.

> ### Box 1. Definitions of symptoms and symptom status in a person with SARS-CoV-2 infections
>
> **Symptoms:** symptoms that a person experiences and reports. We used the authors' definitions. We searched included manuscripts for an explicit statement that the study participant did not report symptoms that they experienced. Some authors defined

"asymptomatic" as an absence of self-reported symptoms. We did not include clinical signs observed or elicited on examination.

**Asymptomatic infection:** a person with laboratory-confirmed SARS-CoV-2 infection, who has no symptoms, according to the authors' report, at the time of first clinical assessment and had no symptoms at the end of follow-up. The end of follow-up was defined as any of the following: virological cure, with 1 or more negative reverse transcription PCR (RT-PCR) test results; follow-up for 14 days or more after the last possible exposure to an index case; follow-up for 7 days or more after the first RT-PCR positive result.

**Presymptomatic:** a person with laboratory-confirmed SARS-CoV-2 infection, who has no symptoms, according to the authors' report, at the time of first clinical assessment, but who developed symptoms by the end of follow-up. The end of follow-up was defined as any of the following: virological cure, with 1 or more negative RT-PCR test results; follow-up for 14 days or more after the last possible exposure to an index case; follow-up for 7 days or more after the first RT-PCR positive result.

## Information sources and search

We conducted the first search on 25 March 2020 and updated it on 20 April 2020, 10 June 2020, 2 February 2021, and, for this update, 6 July 2021. We searched the COVID-19 living evidence database [26], which uses automated workflow processes to: (1) aggregate simultaneous daily searches of 4 electronic databases (Medline, PubMed, Ovid Embase, bioRxiv, and medRxiv), using medical subject headings and free-text keywords for SARS-CoV-2 infection and COVID-19; (2) deduplicate the records; (3) tag records that are preprints; and (4) allow searches of titles and abstracts using Boolean operators. We used the search function to identify studies of asymptomatic or presymptomatic SARS-CoV-2 infection using a search string of medical subject headings and free-text keywords (S1 Text). We also examined articles suggested by experts and the reference lists of retrieved studies. Reports were planned to be updated at 3 monthly intervals, with continuously updated searches.

## Eligibility criteria

We included studies, in any language, of people with SARS-CoV-2 diagnosed by RT-PCR that documented follow-up and symptom status at the beginning and end of follow-up or investigated the contribution to SARS-CoV-2 transmission of asymptomatic or presymptomatic infection. We included contact tracing and outbreak investigations, cohort studies, case-control studies, and mathematical modelling studies. We amended eligibility criteria after the third version of the review [12] in 2 ways. First, we excluded studies that only reported the proportion of presymptomatic SARS-CoV-2 because the settings and methods of these studies were very different and their results were too heterogeneous to summarise [12]. Second, we aimed to reduce the risk of bias from studies with inclusion criteria based mainly on people with symptoms, which would systematically underestimate the proportion of people with asymptomatic infection. We therefore excluded the following study types: case series restricted to people already diagnosed and studies that did not report the number of people tested for SARS-CoV-2, from whom the study population was derived. We also excluded case reports and contact investigations of single individuals or families, and any study without sufficient follow-up (Box 1). Where data from the same study population were reported in multiple records, we extracted data from the most comprehensive report.

## Study selection and data extraction

Reviewers, including crowdsourced-trained volunteers, worked in pairs to screen records using an application programming interface in the electronic data capture system (REDCap, Vanderbilt University, Nashville, Tennessee, United States of America). One reviewer applied eligibility criteria to select studies and a second reviewer verified all included and excluded studies. We reported the process in a flow diagram, adapted for living systematic reviews [27] (S1 Fig). The reviewers determined which of the 3 review questions each study addressed. One reviewer extracted data using a prepiloted extraction form in REDCap and a second reviewer verified the extracted data. For both study selection and data extraction, a third reviewer adjudicated on disagreements that could not be resolved by discussion. We contacted study authors for clarifications. The extracted variables included study design, country and/or region, study setting, population, age, sex, primary outcomes, and length of follow-up (full list of variables in S1 Appendix). We extracted raw numbers of individuals with an outcome of interest and relevant denominators from empirical studies. From statistical and mathematical modelling studies, we extracted proportions and 95% credibility intervals.

The primary outcomes for each review question were (1) the proportion of people with asymptomatic SARS-CoV-2 infection who did not experience symptoms at all during follow-up; (2) secondary attack rate from asymptomatic or presymptomatic index cases, compared with symptomatic cases; (3) model-estimated proportion of SARS-CoV-2 transmission accounted for by people who are asymptomatic or presymptomatic.

## Risk of bias in included studies

After the third version of the review [12], we developed a new tool to assess the risk of bias because the study designs of included studies have changed. In previous versions, we used items from tools to assess case series [28] and the prevalence of mental health disorders [29]. The new tool assessed possible biases in studies of prevalence in general and COVID-19 in particular [4,30]. We developed signalling questions in the domains of selection (2 items), information (3 items), and selective reporting (1 item) biases (S2 Text). For mathematical modelling studies, we used a checklist for assessing relevance and credibility [31]. Two authors independently assessed the risk of bias, using a customised online tool. A third reviewer resolved disagreements.

## Synthesis of the evidence

The data extracted and the code used to display and synthesise the results are publicly available: https://github.com/leonieheron/LSR_Asymp_v5. We used the *metaprop*, *metabin*, and *metafor* functions from the *meta* package (version 4.11–0) [32] and the *ggplot2* package (version 3.3.5) in R (version 3.5.1). The 95% confidence intervals (CIs) for each study were estimated using the Clopper–Pearson method [33]. For review question 1, in contact or outbreak investigations, we subtracted the index cases from the total with SARS-CoV-2 infection, because these people were likely to have been identified because of their symptoms and their inclusion might lead to underestimation of the asymptomatic proportion [16]. For all meta-analyses, we used stratified random effects models. Where a meta-analysis was not done, we present the interquartile range (IQR) and describe heterogeneity visually in forest plots, ordered by study sample size [25]. We calculated the $I^2$ statistic, which is the approximate proportion of between-study variability that is due to heterogeneity other than chance, and $\tau^2$, the between-study variance. We calculated 95% prediction intervals for all summary estimates, to give a likely range of proportions that would have been obtained in hypothetical studies conducted in similar settings [17,19,20]. We did subgroup analyses for prespecified characteristics

of study design, setting and risk of bias, and post hoc analyses according to age group and geographic region, comparing groups using a $\chi^2$ test. We used meta-regression for post hoc analyses examining associations with study size and publication date. To compare our findings with other studies, we extracted the raw data from 5 systematic reviews [11,13–16] and calculated prediction intervals [17]. For review question 2, as a measure of infectiousness, we calculated the secondary attack rate as the number of SARS-CoV-2-infected contacts as a proportion of all close contacts ascertained. For each included study, we compared the secondary attack rate from asymptomatic or presymptomatic index cases with that from symptomatic cases in the same study. If there were no events in a group, we added 0.5 to each cell in the 2 × 2 table. We did not account for potential clustering of contacts because the included studies did not report the number and size of infection clusters consistently. We used the Hartung–Knapp method for random effects meta-analysis to estimate a summary risk ratio (with 95% CI) [34]. For review question 3, we reported the findings descriptively because of large differences between study settings, methods, and results. We did not construct funnel plots to examine bias across studies because their utility in studies reporting on proportions is not clear.

## Results

The searches for studies about asymptomatic or presymptomatic SARS-CoV-2, on 25 March, 20 April, and 10 June 2020 and 2 February and 6 July 2021 resulted in 89, 230, 688, 4,213, and 3,018 records for screening, respectively (S1 Fig). Owing to changes in eligible study designs, this update excludes 67 articles from earlier versions (S1 Table). We included a total of 146 studies addressing 1 or more review questions; 130 empirical studies that estimate the proportion of people with asymptomatic SARS-CoV-2 (summarised in Table 1 and S2 Table) [35–164], 8 studies reporting on secondary attack rates [35,81,131,142,165–168], and 13 mathematical modelling studies [7,165,169–179]. At the time of the search on 6 July 2021, 5 records were preprints. We checked the publication status on 14 March and all remained as preprints [61,62,88,169,171]. The review period from 1 January 2021 onwards includes 52 publications, 3 of which collected data during the period when the alpha variant of concern [85,125,133] had been described and vaccines were being introduced.

### Proportion of people with asymptomatic SARS-CoV-2 infection

The 130 studies reported on 28,426 people with SARS-CoV-2 infection (11,923 defined as having asymptomatic infection) in 42 countries [35–164] (Table 1 and S3 Table). Among all 130 included studies, 88 studies used more than 1 method of follow-up to ascertain asymptomatic status (Table 1 and S2 Table). Only 22 of 130 studies reported the median or mean age [38,47,70,76,77,83,85,95,99,119–121,124,126,128,133,134,139,143,146,152,164] and only 5 studies included children only [65,67,110,115,118]. Only 31 studies reported the sex of people with asymptomatic SARS-CoV-2 (Table 1 and S2 Table) [38,47,51,53,70,71,75,76,83,85,95,99,107, 119–122,124,126,128,133,134,139,143,146,147,150,153,158,162,164]. The types of included studies changed across the 5 versions of the review. In the first version [23], 6 of 9 studies were contact tracing investigations of single-family clusters. In this version, 2 main types of study design generated the study populations of people with SARS-CoV-2: contact tracing or outbreak investigation methods were used to identify and test potentially infected contacts (46 studies, referred to as contact and outbreak investigations); and studies that involved screening of a defined group of people in settings in the community, institutions, such as long-term care facilities, or occupational groups (84 studies, referred to as screening studies).

Between-study heterogeneity was considerable, so we did not estimate a mean proportion of asymptomatic infections overall, or for screening studies (Fig 1). The IQR of estimates for

**Table 1. Summary of characteristics of studies reporting on proportion of asymptomatic SARS-CoV-2 infections (review question 1).**

| | Study design and setting | | | | | All studies |
|---|---|---|---|---|---|---|
| | Contact investigation | Outbreak investigation | Screening of defined population | | | |
| | | | Community | Institutional | Occupational | |
| Total studies[a], *n* | 13 | 33 | 23 | 43 | 18 | 130 |
| **Publication date** | | | | | | |
| January 2020–June 2020 | 5 | 9 | 3 | 3 | 4 | 24 |
| July 2020–December 2020 | 5 | 17 | 11 | 17 | 6 | 56 |
| January 2021 onwards | 3 | 7 | 9 | 23 | 8 | 50 |
| **Region[b]** | | | | | | |
| Africa | 0 | 2 | 2 | 1 | 1 | 6 |
| Americas | 5 | 10 | 4 | 19 | 7 | 45 |
| Southeast Asia | 0 | 3 | 2 | 2 | 1 | 7 |
| Europe | 2 | 13 | 7 | 18 | 5 | 45 |
| Eastern Mediterranean | 0 | 0 | 3 | 2 | 2 | 8 |
| Western Pacific | 6 | 6 | 5 | 1 | 2 | 19 |
| **Follow-up method[c]** | | | | | | |
| 14 days after last possible exposure | 7 | 11 | 2 | 3 | 4 | 27 |
| ≥7 days after diagnosis | 11 | 27 | 19 | 35 | 16 | 108 |
| Until negative RT-PCR result | 2 | 4 | 9 | 12 | 6 | 33 |
| Two or more follow-up methods | 8 | 21 | 17 | 28 | 14 | 88 |
| **Age range of study participants** | | | | | | |
| Children (<18 years) | 1 | 1 | 0 | 3 | 0 | 5 |
| Adults (18–65 years) | 3 | 10 | 9 | 16 | 14 | 52 |
| Older adults (>65 years) | 0 | 7 | 0 | 6 | 0 | 13 |
| All ages | 7 | 14 | 10 | 15 | 2 | 48 |
| No information about age | 2 | 1 | 4 | 3 | 2 | 12 |
| Total with SARS-CoV-2 infection, *n* | 1,076 | 4,910 | 10,652 | 8,921 | 2,867 | 28,426 |
| Asymptomatic SARS-CoV-2 infections | 264 | 1,409 | 6,007 | 3,658 | 585 | 11,923 |
| **Sex of asymptomatic cases[d]** | | | | | | |
| Male | 1 | 33 | 1,419 | 30 | 16 | 1,499 |
| Female | 0 | 32 | 589 | 313 | 26 | 960 |

RT-PCR, reverse transcription PCR; SARS-CoV-2, Severe Acute Respiratory Syndrome Coronavirus 2.

[a]S2 Table reports the characteristics of each study included.

[b]World Health Organization regions.

[c]Studies could have more than 1 method of follow-up (S2 Table).

[d]Ninety-nine studies did not report the gender of asymptomatic cases.

all 130 included studies (141 clusters) was 14% to 50% (prediction interval 2% to 90%). In 46 studies enrolling people found through contact or outbreak investigations, for example, in long-term care facilities, in aeroplanes, or on cruise ships, we estimated a summary estimate for the proportion asymptomatic (19%, 95% CI 15% to 25%, prediction interval 2% to 70%, IQR 8% to 37%). The estimated proportions of asymptomatic SARS-CoV-2 infections were similar in studies of contact investigations (16%, 95% CI 9% to 27%, IQR 8% to 38%, 13 studies) and outbreak investigations (20%, 95% CI 15% to 28%, IQR 8% to 38%, 33 studies) (S2 Fig).

In 84 screening studies, the IQR for estimates from individual studies was 20% to 65% and the prediction interval was 4% to 94% [41–48,50–54,56,58–63,67,69–71,73,74,79,83,86,88–97,99–101,103–105,107,109,110,114–117,119,120,122,124–126,128,130,133–138,143–145,147,150–156,158–164]. The ranges of estimates were similar in 3 settings in which

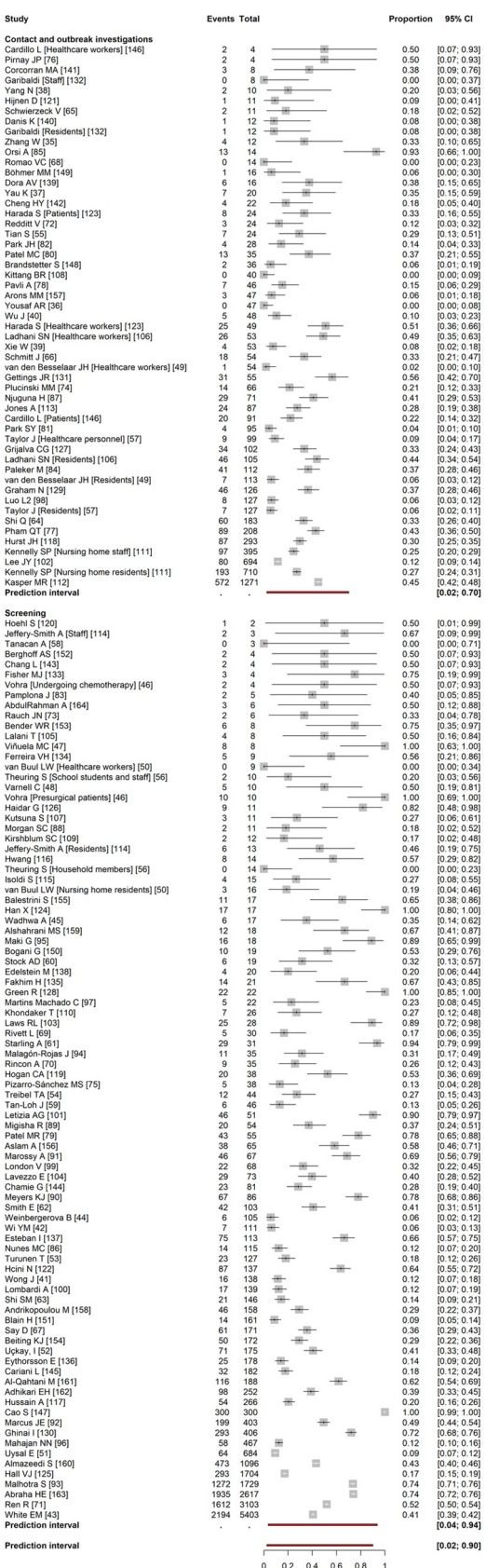

**Fig 1. Forest plot of proportion of people with asymptomatic SARS-CoV-2 infection, stratified by study design.** In contact and outbreak investigations, the summary estimate for meta-analysis was 19% (15%–25%) and the IQR was 8%–37%. In screening studies, the IQR was 20%–65%. Across all studies, the IQR was 14%–50%. The x-axis displays proportions. Where more than 1 cluster was reported, clusters are annotated with [cluster identity]. The IQR is given below the individual study estimates. The red bar shows the prediction interval. CI, confidence interval; IQR, interquartile range; SARS-CoV-2, Severe Acute Respiratory Syndrome Coronavirus 2.

screening studies were conducted; people in a community setting (23 studies, IQR 20% to 53%, prediction interval 2% to 96%), institutional settings such as nursing homes (43 studies, IQR 26% to 67%, prediction interval 5% to 93%), and occupational settings such as among groups of healthcare workers (18 studies, IQR 17% to 64%, prediction interval 3% to 95%) (S2 Fig).

Three studies had data collection periods from 1 January 2021. In 2 nursing home outbreaks, 3/4 SARS-CoV-2 infections in partially vaccinated residents [133] and 13/14 infections (with the alpha variant) [85] in full vaccinated residents were asymptomatic. One study among healthcare workers did not report on symptom status according to vaccination or variant of concern but found that 76/155 (49%) with reinfection compared with 273/1,704 (17%) with primary infection were asymptomatic [125].

**Risk of bias in individual studies.** There were risks of bias in all types of empirical studies (S3 Fig). In prespecified subgroup analyses according to risk of bias domains (S4 Table), statistical heterogeneity remained very high ($I^2 \geq 84\%$) and the prediction intervals remained wide. In screening studies, the summary proportion in was lower in studies judged to be at low risk of information bias in the assessment of symptoms (29%, 95% CI 20% to 42%) than in studies at unclear or high risk of bias (47%, 95% CI 37% to 57%) ($p$ = 0.03, test for subgroup differences). For all other domains, estimates of the proportion asymptomatic were not associated with the assessment of the risk of bias.

**Factors associated with proportion of asymptomatic SARS-CoV-2.** Table 2 shows analyses for factors prespecified in the protocol and post hoc (Table 2). Study design and setting was prespecified and explained 16% of the variance in heterogeneity; estimates ranged from 16% (8% to 29%) for contact investigations to 45% (35% to 56%) for screening studies in institutional settings (S2 Fig). The date of publication was associated with the estimate of the proportion asymptomatic (S4 Fig), increasing in more recent publications (p 0.03), although this only explained 4% of the variance in heterogeneity. There was some evidence of variability in different world regions ($p$ = 0.06), explaining 9% of the heterogeneity. Sample size and age range of the study participants did not appear to influence the estimated asymptomatic proportion. In 3 systematic reviews that we reanalysed, prediction intervals were: 1% to 83% (241 studies [14]); 4% to 97% (95 studies [15]); and 3% to 89% (170 studies [16]). $I^2$ values were between 94% and 99% (S2 Appendix).

## Infectiousness of people with asymptomatic or presymptomatic SARS-CoV-2

Eight studies provided data to calculate and to compare secondary attack rates by symptom status of the index case (Fig 2) [35,81,131,142,165–168]. Seven studies compared the secondary attack rate from asymptomatic with symptomatic index cases (summary risk ratio 0.32 (95% CI 0.16 to 0.64, prediction interval 0.11 to 0.95) [81,131,142,165–167]. One study compared asymptomatic with presymptomatic index cases (summary risk ratio 0.19, 95% CI 0.02 to 1.46) [35] and 4 studies compared presymptomatic with symptomatic index cases (summary risk ratio 1.00 (95% CI 0.37 to 2.71, prediction interval 0.11 to 9.12) [81,142,166,167]. The risk of

**Table 2. Summary of findings of subgroup and meta-regression analyses of factors associated with the proportion of asymptomatic SARS-CoV-2 infections.**

| Variable | Clusters[a] | Proportion at the reference value (95% CI) | $\tau^{2b}$ | p-Value (subgroup difference/ intercept) | Heterogeneity variance explained[c] |
|---|---|---|---|---|---|
| Reference | 141 | 0.32 (0.27–0.38) | 2.19 | - | - |
| **Study design and setting[d]** | | | | | |
| Contact investigation | 13 | 0.16 (0.08; 0.29) | 1.87 | <0.001 | 16% |
| Outbreak investigation | 40 | 0.20 (0.14; 0.28) | | | |
| Screening: community | 24 | 0.39 (0.26; 0.53) | | | |
| Screening: institutional | 46 | 0.45 (0.35; 0.56) | | | |
| Screening: occupational | 18 | 0.41 (0.26; 0.59) | | | |
| **Age group[e]** | | | | | |
| All ages | 57 | 0.28 (0.21–0.37) | 2.11 | 0.38 | 4% |
| Adults (18–65 years) | 52 | 0.36 (0.27–0.47) | | | |
| Older adults (>65 years) only | 14 | 0.25 (0.13–0.44) | | | |
| Children (<18 years) only | 5 | 0.27 (0.09–0.58) | | | |
| Not reported | 13 | 0.45 (0.27–0.66) | | | |
| **Geographic region[e,f]** | | | | | |
| Americas | 47 | 0.37 (0.27–0.47) | 2.00 | 0.06 | 9% |
| Europe | 52 | 0.24 (0.17–0.33) | | | |
| Western Pacific | 20 | 0.35 (0.21–0.51) | | | |
| Southeast Asia | 8 | 0.22 (0.09–0.43) | | | |
| East Mediterranean | 8 | 0.59 (0.33–0.81) | | | |
| Africa | 6 | 0.47 (0.22–0.74) | | | |
| **Selection bias[a,d]** | | | | | |
| Low risk | 54 | 0.34 (0.25–0.44) | 2.19 | 0.67 | 0% |
| Unclear/high risk | 87 | 0.31 (0.25–0.39) | | | |
| **Information bias, assessment of symptoms defining status[a,d]** | | | | | |
| Low risk | 33 | 0.25 (0.16–0.36) | 2.15 | 0.14 | 2% |
| Unclear/high risk | 108 | 0.35 (0.28–0.42) | | | |
| **Information bias, misclassification based on follow-up[a,d]** | | | | | |
| Low risk | 107 | 0.32 (0.25–0.38) | 2.19 | 0.65 | 0% |
| Unclear/high risk | 34 | 0.35 (0.24–0.47) | | | |
| **Selective reporting bias[a,d]** | | | | | |
| Low risk | 126 | 0.33 (0.27–0.40) | 2.17 | 0.37 | 1% |
| Unclear/high risk | 15 | 0.25 (0.13–0.43) | | | |
| **Sample size[g]** | | | | | |
| Proportion at 50 | - | 0.29 (0.14–0.51) | 1.91 | 0.06 | 13% |
| Proportion at 120 | - | 0.31 (0.24–0.39) | | | |
| Proportion at 200 | - | 0.32 (0.24–0.40) | | | |
| **Publication date** | | | | | |
| Reference (first date, 19 Feb 2020) | - | 0.21 (0.13–0.32) | 2.09 | <0.001 | 4% |
| Coefficient | - | 0.50 (0.50–0.50) | | | |

CI, confidence interval; SARS-CoV-2, Severe Acute Respiratory Syndrome Coronavirus 2.

[a]Total number of studies, 130; independent within-study clusters counted individually;.

[b]Common heterogeneity parameter estimated within each subgroup.

[c]Formula for proportion of heterogeneity variance explained, $\frac{\tau^2_{unadjusted} - \tau^2_{adjusted}}{\tau^2_{unadjusted}}$.

[d]Prespecified analysis in review protocol.

[e]Subgroup analysis not specified in review protocol.

[f]World Health Organization regions.

[g]Prevalence estimated using the meta-regression model for the approximate values of the median ($n = 46$), the mean ($n = 202$), and the third quartile ($n = 126$) of study sample sizes.

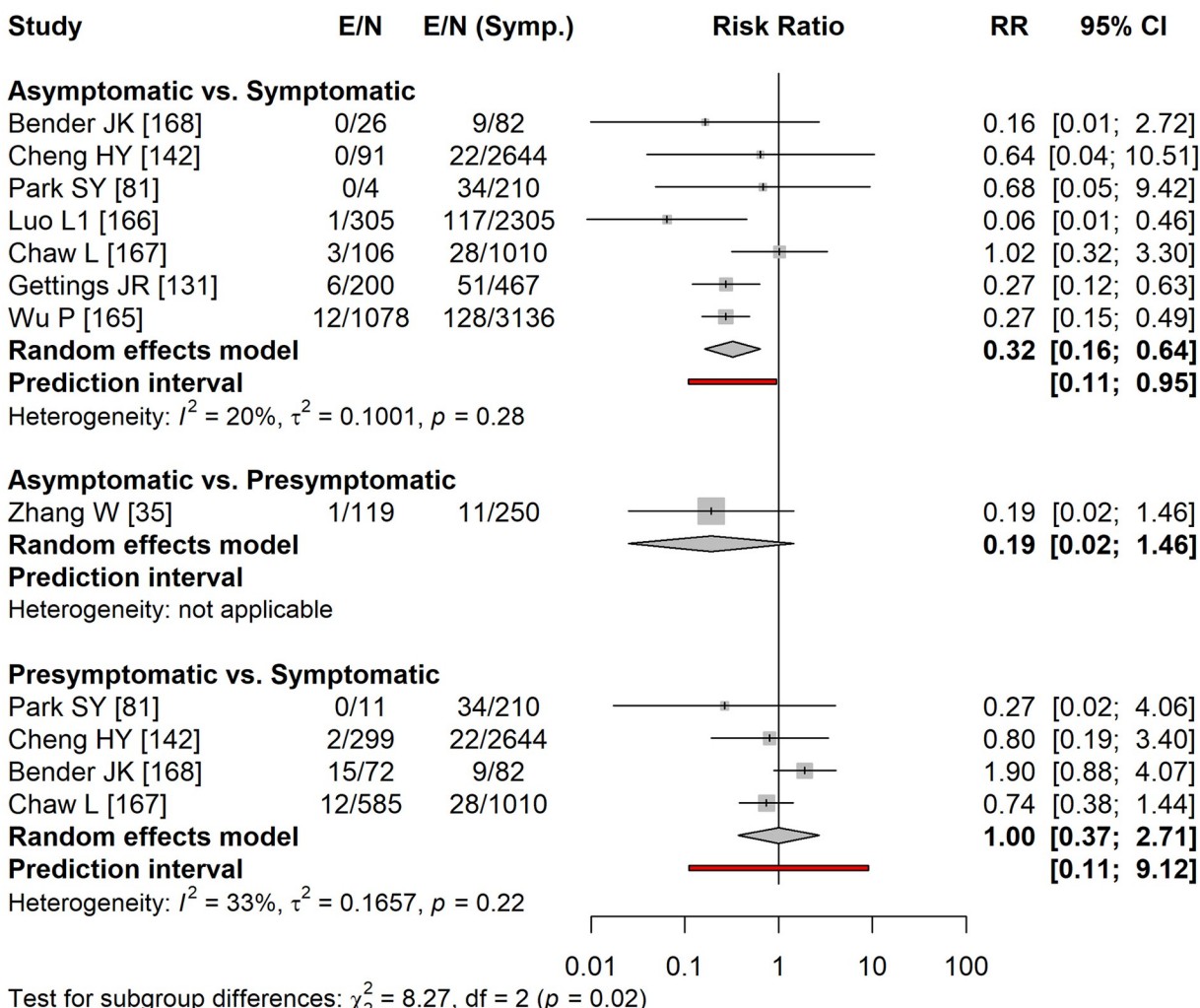

**Fig 2. Forest plot of the secondary attack rate of SARS-CoV-2 infections, comparing infections in contacts of asymptomatic and presymptomatic index cases with infections in contacts of symptomatic cases.** The RR is on a logarithmic scale. The diamonds show the summary estimate and its 95% CI. The red bar shows the prediction interval. CI, confidence interval; E, number of secondary transmission events; N, number of close contacts; RR, risk ratio; SARS-CoV-2, Severe Acute Respiratory Syndrome Coronavirus 2; Symp., symptomatic individuals.

information bias, specifically in symptom assessment, was judged to be high or unclear in 5 of the 8 studies included (S3 Fig).

## Contribution of asymptomatic and presymptomatic infection to SARS-CoV-2 transmission

We included 13 mathematical modelling studies (Fig 3 and S5 Table) [7,165,169–179]. The models in 9 studies were informed by analyses of data from contact investigations in China, South Korea, Singapore, and from an outbreak on the *Diamond Princess* cruise ship, using data to estimate the serial interval or generation time [7,165,170,172,173,176–179]. In 3 studies, the authors did not analyse any original data sources [169,174,175].

Estimates of the contributions of both asymptomatic and presymptomatic infections SARS-CoV-2 transmission were very heterogeneous. For asymptomatic SARS-CoV-2 infection, 4 studies contributed 5 estimates [7,171,175,177]. In 3 studies, the estimates suggested a

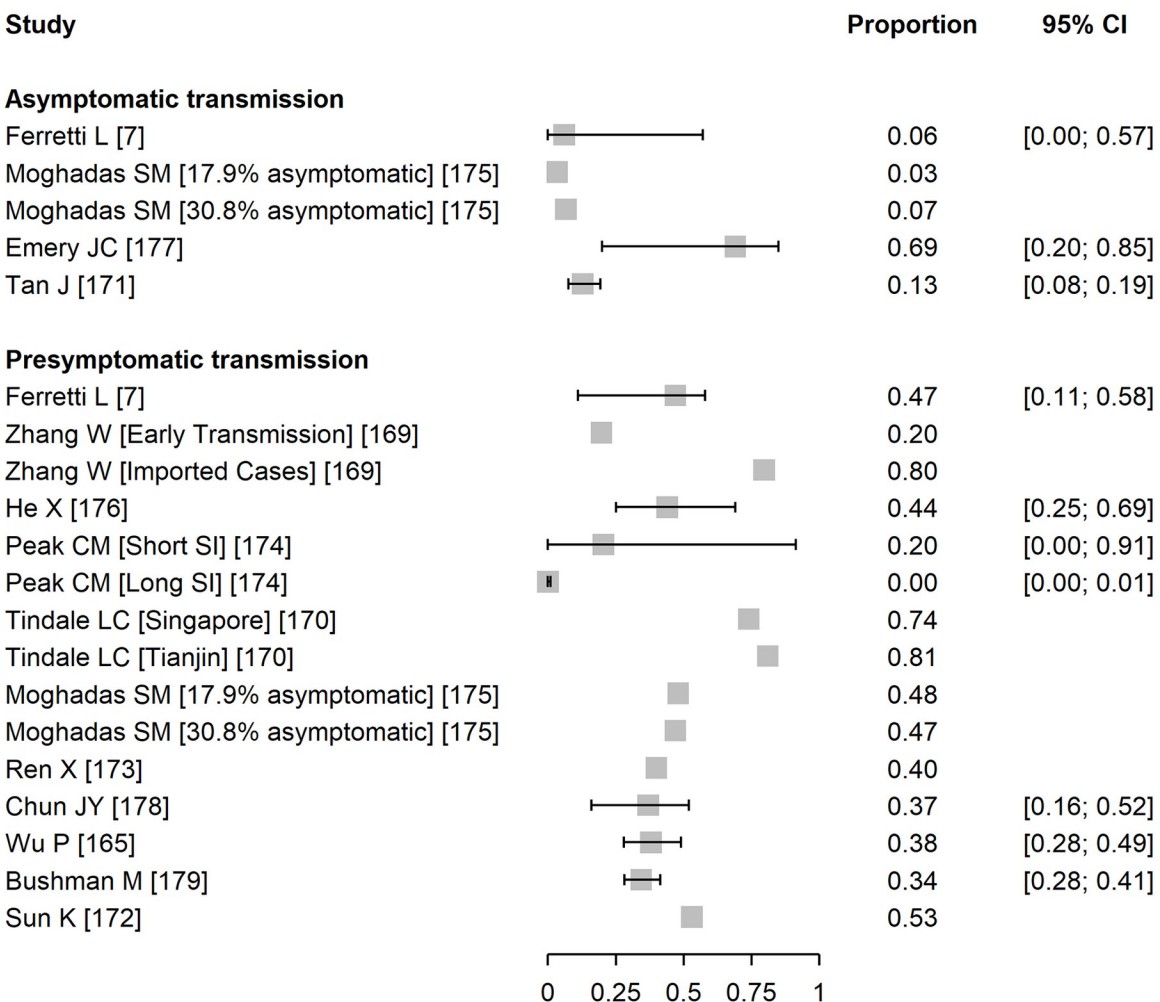

**Fig 3. Forest plot of proportion of SARS-CoV-2 infection resulting from asymptomatic or presymptomatic transmission.** For studies that report outcomes in multiple settings, these are annotated in brackets. CI, confidence interval; SI, serial interval; SARS-CoV-2, Severe Acute Respiratory Syndrome Coronavirus 2.

contribution to SARS-CoV-2 transmission of asymptomatic infection of less than 15%. One study estimated a higher proportion (69%, 95% CI 20% to 85%) with a wide credibility interval [177] (Fig 3). The estimates have large uncertainty intervals, and the disparate predictions result from differences in the proportion of asymptomatic infections and relative infectiousness of asymptomatic infection.

We included 12 studies providing 16 estimates of the contribution of presymptomatic transmission [7,165,169,170,172–179]. The models examined a range of epidemic settings and used different assumptions about the durations and distributions of infection parameters such as incubation period, generation time, and serial interval (S5 Table). In 7 studies, point estimates for the estimated contribution of presymptomatic infection to all SARS-CoV-2 transmission in at least 1 reported scenario were 40% or greater [7,169,170,172,173,175,176] (Fig 3). In 1 study that estimated a contribution of <1% [174], the model-fitted serial interval was longer than observed in empirical studies. The credibility of most modelling studies was limited

by the absence of external validation and of uncertainty intervals for the estimates cited (S5 Fig). The estimates from studies that relied on data from different published sources that might not have been compatible were assessed as providing low quality evidence (S5 Table).

## Discussion

### Summary of main findings

(1) For all 130 included studies, the IQR for the proportion of asymptomatic SARS-CoV-2 was 14% to 50%, prediction interval 2% to 90%) and for 84 studies based on screening of defined populations, IQR 20% to 65% (prediction interval 4% to 94%). In 46 studies that identified participants through contact tracing of index cases and outbreak investigations, the summary proportion from meta-analysis was 19% (95% CI 15% to 25%, prediction interval 2% to 70%). (2) The risk ratio for the secondary attack rate from asymptomatic compared with symptomatic infections was 0.32 (95% CI 0.16 to 0.64, prediction interval 0.11 to 0.95) and for presymptomatic infections compared with symptomatic infection was 1.00 (95% CI 0.37 to 2.71, prediction interval 0.11 to 9.12). (3) In mathematical modelling studies, estimated proportions of all SARS-CoV-2 infections that result from transmission from asymptomatic individuals were mostly below 15%, and from presymptomatic individuals mostly higher than 40%. Evidence about asymptomatic infections caused by variants of concern, or with immunity following infection or vaccination, is limited.

### Strengths and weaknesses of the living systematic review methods

A strength of the methodology of this review is the transparent reporting, with openly available data and changes over different versions reported in the protocol. Our inclusion criteria attempted to reduce risks of bias and we developed a new tool to address potential biases in the studies included in this review. In contact investigations, we subtracted index cases from the total number of people with SARS-CoV-2 to avoid underestimation of the proportion asymptomatic [16]. We examined heterogeneity in detail and, as a result of the wide prediction interval, we chose not to report an overall summary estimate [18,25]. A limitation of the methods for this living systematic review is that this update only includes published studies up to 6 July 2021. This covers the period when vaccines started to be rolled out and the alpha variant of concern became dominant in high-income countries. Although we made extensive efforts to comply with the planned 3 monthly updates, with weekly searches and a continuous process of screening, data extraction and risk of bias assessment, the pace of publications about SARS-CoV-2 exceeds the capacity of our crowd of reviewers [10,26]. Our decision to include pre-prints compensates for some of the delay because these articles appear sooner than peer-reviewed publications. In reviews of observational epidemiological studies, search terms are broad so the number of studies that needs to be screened is high, but the yield of included studies is low. The 4 databases that we searched are not comprehensive, but they cover the majority of publications and we do not believe that we have missed studies that would change our conclusions. We have also not considered the possible impact of false negative RT-PCR results, which might be more likely to occur in asymptomatic infections [180] and would underestimate the proportion of asymptomatic infections [181].

### Comparison with other reviews and interpretation

The type of studies that provide estimates of the proportion of asymptomatic SARS-CoV-2 infections and heterogeneity between them has changed over the course of the pandemic. In our living systematic review, the prediction interval has widened from 23% to 37% in studies

published up to 25 March 2020 [23], to 3% to 67% up to June 2020 [12], 2% to 89% up to 2 February 2021 [21] and remains at 2% to 90% up to 6 July. We found 3 systematic reviews, in which authors reported restriction to studies with adequate follow-up (S2 Appendix) [11,13,16]. In 2 of the reviews, authors also applied criteria to reduce the risks of selection bias, with summary estimates of 18% (95% CI 9% to 26%, $I^2$ 84%, 9 studies) [13] and 23% (95% CI 16% to 30%, $I^2$ 92%, 21 studies) [11]. In both reviews, with studies published up to mid-2020, many included studies used designs that we defined as contact or outbreak investigations (Fig 1, S2 Table, and S2 Fig). Sah and colleagues reviewed studies published up to April 2021 and their subgroup estimate from studies in long-term care facilities, which include many outbreak investigations, was 17.8%, 95% CI 9.7% to 30.3%, 15 studies [16]. The summary estimates from all these reviews are compatible with our estimate from 46 studies in similar settings (19%, 95% CI 15% to 25%, prediction interval 2% to 70%, $I^2$ 90%) (Fig 1).

It may not be possible to obtain a single summary estimate from published literature of the proportion of persistently asymptomatic SARS-CoV-2 infection. Estimates from meta-analysis might be precise, but are likely to be unreliable owing to unacceptably high levels of heterogeneity. In 3 large systematic reviews, overall estimates had narrow confidence intervals [14–16], but $I^2$ values were 94% to 99% and prediction intervals, which show the extent of all between-study variability were not reported [17]. The prediction intervals that we calculated extended more or less from zero to 100% (S2 Appendix), making differences in estimates between these studies hard to interpret. We expected this update to our living systematic review to provide a more precise and less heterogeneous estimate of the proportion of people with asymptomatic SARS-CoV-2 than in the previous version [12]. In particular, we expected that studies that detect SARS-CoV-2 through screening of defined populations and follow up of those infected would be less affected by biases in study methodology [30] and would provide a more accurate estimate of persistently asymptomatic SARS-CoV-2, which should be influenced mainly by properties of the virus and the host response to infection [182]. Study design was the factor that explained the largest proportion of variability in this review (S2 Appendix). Information bias, resulting from the way in which asymptomatic status is determined, was the factor most strongly associated with the estimated proportion of asymptomatic infection in screening studies (Table 2). Studies based on contact and outbreak investigations might obtain more detailed data about symptoms, resulting in lower estimates of the proportion that is classified as asymptomatic. Symptom report also differs between different groups of study participants, even within the same study, and could also contribute to heterogeneity [183]. Age might play a role as children appear more likely than adults to have an asymptomatic course of infection, but age was poorly reported and could not be examined in detail (Tables 1 and 2).

The analysis of secondary attack rates in this update now provides strong evidence of lower infectiousness of people with asymptomatic than symptomatic infection (Fig 2). The difference in secondary attack rates between asymptomatic and symptomatic index cases in our meta-analysis is smaller and less biased than in systematic reviews that analyse groups of studies reporting asymptomatic index cases and of symptomatic cases separately [182,184]. In meta-analyses of 2 proportions, the direct comparison within studies reduces heterogeneity and is less biased [34]. Since SARS-CoV-2 can be transmitted a few days before the onset of symptoms [185], presymptomatic transmission likely contributes substantially to overall SARS-CoV-2 epidemics. If both the proportion and transmissibility of asymptomatic infection are relatively low, people with asymptomatic SARS-CoV-2 infection should account for a smaller proportion of overall transmission than presymptomatic individuals. This is consistent with the findings of modelling studies in our review, although the absence of descriptions of the epidemic context in many studies made it difficult to compare findings across studies.

## Implications and unanswered questions

This living systematic review shows the challenges of synthesising evidence from observational epidemiological studies. Heterogeneity in systematic reviews of prevalence is a recognised challenge [34,186]. Methodological guidance to refrain from meta-analysis, and to report prediction intervals, when the variability between studies is extreme is often ignored in favour of summary estimates, which are easy to cite [18,20]. Part of the heterogeneity in our review arises from the fact that many studies were not designed to estimate the proportion of asymptomatic SARS-CoV-2 infection. The incomplete descriptions of inclusion criteria, response rates, follow-up, and of definitions of symptom status [1] made it difficult to assess the risks of bias and to investigate their contribution to between-study heterogeneity.

The finding that, in studies of contact and outbreak investigations, a substantial minority of people with SARS-CoV-2 infection remains asymptomatic throughout the course of infection, and that almost half of all transmission might occur before symptoms develop has already had implications for prevention. When SARS-CoV-2 community transmission levels are high, physical distancing measures and mask-wearing need to be sustained to prevent transmission from close contact with people with asymptomatic and presymptomatic infection. Integration of evidence from epidemiological, clinical, and laboratory studies will help to clarify the relative infectiousness of asymptomatic SARS-CoV-2. Studies using viral culture as well as RNA detection are needed since RT-PCR defined viral loads appear to be broadly similar in asymptomatic and symptomatic people [180,187].

Determining the viral dynamics and full clinical spectrum of infection with variants of concern is important. Variants classed as omicron differ substantially from all earlier SARS-CoV-2 variants, with high infectiousness and immune evasion [188], and viral characteristics and immunity could influence the occurrence of asymptomatic infection. Studies published in early 2022 are already reporting a wide range of estimates of asymptomatic omicron infection. In India, from the date of emergence of the omicron variant, 24 November 2021 to 4 January 2022, authors reported a high proportion of asymptomatic omicron variant infections (56.7% of 291) but did not report any follow-up and >80% of participants had been vaccinated [189]. In contrast, authors of a cohort study of an outbreak of omicron SARS-CoV-2 in Norway, found only 1 of 81 infections in a highly vaccinated group was asymptomatic after 10 days of follow-up [190]. There are increasing challenges for studies relying on routine health service or surveillance data; in many jurisdictions, indications for routine testing are being reduced, which will make selection biases more likely, and mandated quarantine and isolation periods for people with diagnosed SARS-CoV-2 infection are being reduced, which will increase information biases in the ascertainment of persistent asymptomatic status. Researchers need to design studies to address this specific research question for each variant of concern, taking into account vaccination status and prior infection. There are ongoing prospective studies that collect appropriate data [125], for which improved reporting could address the requirements for assessing asymptomatic infection status fully, but ongoing funding for these studies is not secure [191]. Without prospective longitudinal studies with methods that minimise selection and measurement biases, further updates to this living systematic review are unlikely to provide a reliable summary estimate of the proportion of asymptomatic infections caused by SARS-CoV-2.

## Supporting information

**S1 PRISMA Checklist.**
(PDF)

**S1 Text. Search strings.**
(PDF)

**S2 Text. Risk of bias tool.**
(PDF)

**S1 Appendix. Data extraction forms.**
(PDF)

**S2 Appendix. Analysis of other systematic reviews on asymptomatic SARS-CoV-2.**
(PDF)

**S1 Table. Studies included in version 3 and excluded in versions 4 and 5 of the living systematic review.**
(PDF)

**S2 Table. Characteristics of studies reporting on proportion of asymptomatic SARS-CoV-2 infections (review question 1 and question 2).** F, female; IQR, interquartile range; M, male; NR, not reported; SARS-CoV-2, Severe Acute Respiratory Syndrome Coronavirus 2.
(PDF)

**S3 Table. Location of studies contributing data to review question 1.**
(PDF)

**S4 Table. Subgroup analysis according to risk of bias.**
(PDF)

**S5 Table. Characteristics of mathematical modelling studies.** CI, confidence interval; NPI, non-pharmaceutical intervention; SARS-CoV-2, Severe Acute Respiratory Syndrome Coronavirus 2.
(PDF)

**S1 Fig. Flowchart of identified, excluded, and included records as of 6 July 2021.**
(PDF)

**S2 Fig. Forest plot of proportion of people with asymptomatic SARS-CoV-2 infection, stratified by study design.**
(PDF)

**S3 Fig. Risk of bias assessment of studies in question 1 and 2.**
(PDF)

**S4 Fig. Forest plot of proportion of people with asymptomatic SARS-CoV-2 infection by date of publication.**
(PDF)

**S5 Fig. Assessment of credibility of mathematical modelling studies.** NA, not applicable; NEI: not enough information; NR: not reported; PY: partially yes.
(PDF)

## Acknowledgments

We are very grateful to Dianne Egli-Gany, Michel J. Counotte, and Stefanie Hossmann who were part of our original core team of reviewers and contributed to the first 3 versions of this living systematic review. Without their contributions, we would not have been able to complete the fourth and fifth versions.

We would like to thank Alexander Holloway for his valuable help with programming the automated workflows in our living systematic review processes.

This work reflects only the authors' view. The European Commission is not responsible for any use that may be made of the information it contains.

## Author Contributions

**Conceptualization:** Diana Buitrago-Garcia, Aziz Mert Ipekci, Leonie Heron, Nicola Low.

**Data curation:** Diana Buitrago-Garcia, Aziz Mert Ipekci, Leonie Heron, Hira Imeri, Lucia Araujo-Chaveron, Ingrid Arevalo-Rodriguez, Agustín Ciapponi, Muge Cevik, Muhammad Irfanul Alam, Kaspar Meili, Eric A. Meyerowitz, Nirmala Prajapati, Xueting Qiu, Aaron Richterman, William Gildardo Robles-Rodriguez, Shabnam Thapa, Ivan Zhelyazkov.

**Formal analysis:** Leonie Heron, Georgia Salanti.

**Funding acquisition:** Diana Buitrago-Garcia, Nicola Low.

**Investigation:** Diana Buitrago-Garcia, Aziz Mert Ipekci, Leonie Heron, Ingrid Arevalo-Rodriguez, Agustín Ciapponi, Anthony Hauser, Eric A. Meyerowitz, Nirmala Prajapati, Aaron Richterman, William Gildardo Robles-Rodriguez, Nicola Low.

**Methodology:** Diana Buitrago-Garcia, Aziz Mert Ipekci, Leonie Heron, Anthony Hauser, Georgia Salanti, Nicola Low.

**Project administration:** Diana Buitrago-Garcia, Aziz Mert Ipekci, Leonie Heron, Hira Imeri.

**Software:** Leonie Heron, Hira Imeri.

**Supervision:** Georgia Salanti, Nicola Low.

**Validation:** Diana Buitrago-Garcia, Aziz Mert Ipekci, Leonie Heron, Hira Imeri, Lucia Araujo-Chaveron, Ingrid Arevalo-Rodriguez, Agustín Ciapponi, Muge Cevik, Anthony Hauser, Muhammad Irfanul Alam, Kaspar Meili, Eric A. Meyerowitz, Nirmala Prajapati, Xueting Qiu, Aaron Richterman, William Gildardo Robles-Rodriguez, Shabnam Thapa, Ivan Zhelyazkov, Georgia Salanti, Nicola Low.

**Visualization:** Leonie Heron, Georgia Salanti, Nicola Low.

**Writing – original draft:** Diana Buitrago-Garcia, Aziz Mert Ipekci, Leonie Heron, Nicola Low.

**Writing – review & editing:** Diana Buitrago-Garcia, Aziz Mert Ipekci, Leonie Heron, Hira Imeri, Lucia Araujo-Chaveron, Ingrid Arevalo-Rodriguez, Agustín Ciapponi, Muge Cevik, Anthony Hauser, Muhammad Irfanul Alam, Kaspar Meili, Eric A. Meyerowitz, Nirmala Prajapati, Xueting Qiu, Aaron Richterman, William Gildardo Robles-Rodriguez, Shabnam Thapa, Ivan Zhelyazkov, Georgia Salanti, Nicola Low.

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
