## [Editor Report · Decision Letter 0]

24 Jan 2022

Dear Dr Low, 

Thank you for submitting your manuscript entitled "Occurrence and transmission potential of asymptomatic and presymptomatic SARS-CoV-2 infections: a living systematic review and meta-analysis" for consideration at PLOS Medicine.

Your manuscript has now been evaluated by the PLOS Medicine editorial staff and I am writing to let you know that we would like to send your submission out for external assessment.

However, before we can send your manuscript for assessment, we need you to complete your submission by providing the metadata that are required. To this end, please login to Editorial Manager where you will find the paper in the 'Submissions Needing Revisions' folder on your homepage. Please click 'Revise Submission' from the Action Links and complete all additional questions in the submission questionnaire.

Please re-submit your manuscript within two working days, i.e. by Jan 26 2022 11:59PM.

Once your full submission is complete, your paper will undergo a series of checks in preparation for full assessment. 

Kind regards,

Richard Turner, PhD

plosmedicine@plos.org

---

## [Decision Letter · Decision Letter 1]

17 Feb 2022

Dear Dr. Low,

Thank you very much for submitting your manuscript "Occurrence and transmission potential of asymptomatic and presymptomatic SARS-CoV-2 infections: a living systematic review and meta-analysis" (PMEDICINE-D-22-00260R1) for consideration at PLOS Medicine. 

Your paper was discussed among the editors and sent to independent reviewers, including a statistical reviewer. The reviews are appended at the bottom of this email and any accompanying reviewer attachments can be seen via the link below:

[LINK]

In light of these reviews, we will not be able to accept the manuscript for publication in the journal in its current form, but we would like to invite you to submit a revised version that addresses the reviewers' and editors' comments fully. You will recognize that we cannot make a decision about publication until we have seen the revised manuscript and your response, and we expect to seek re-review by one or more of the reviewers. 

We hope to receive your revised manuscript by Mar 10 2022 11:59PM. Please email us (plosmedicine@plos.org) if you have any questions or concerns.

Please let me know if you have any questions, and we look forward to receiving your revised manuscript. 

Sincerely,

Richard Turner, PhD

Senior editor, PLOS Medicine

rturner@plos.org

As requested by one or more of our reviewers, please update the search to the most recent date possible.

After the abstract, please add a new and accessible "author summary" section in non-identical prose. You may find it helpful to consult one or two recent research papers published in PLOS Medicine to get a sense of the preferred style. 

At line 61, we suggest "we conducted an update ..." or similar. 

Please remove the information on funding and competing interests from the end of the main text. In the event of publication, this will appear in the article metadata, via entries in the submission form. 

Noting one entry in table 2, please quote exact p values or "p<0.001" throughout, unless there are specific reasons to do otherwise. 

Throughout the text, please remove spaces from within the square brackets for reference call-outs (e.g., "... more [2,3].").

In the reference list, please add "[preprint]" to all preprint citations, noting reference 33 and others.

Noting reference 10, please use the journal name abbreviation "PLoS Med."; and for reference 57 "PLoS ONE".

Noting reference 95 and others, please ensure that all references have full access details. 

Thank you very much for including the PRISMA checklist. Please adapt this so that individual items are referred to by section (e.g., "Methods") and paragraph numbers, not by line or page numbers as these generally change in the event of publication. 

Please remove the attached earlier version of the systematic review. 

Comments from the reviewers:

*** Reviewer #1: 

This is an excellent review of an important set of questions 

The review is stated as being a "living" review, but the literature was only searched up to February 2021. So what is the definition of a living review, according to these investigators. A review of publications up to one year ago does not seem to fit with definitions of living reviews.

While it is understandable that the production of literature and resources to conduct the review process is limiting the ability to keep up to date, it would be good to think about restrictions that could be made to streamline this process and allow for more recent updating (eg dropping pre-prints, increasing minimal sample size for eligibility, restricting to particular study designs, excluding based on certain quality criteria). The WHO Covid-19 database is a one-stop shop that could be used instead of searching 4 separate databases

More updated information may also contribute information about the predominant variant in circulation at the time of the study.

The Comparison with other reviews is helpful and could be summarised graphically. 

*** Reviewer #2:

This is a thoughtful and well-written update to a living systematic review and meta-analysis examining the prevalence and roles of asymptomatic/presymptomatic infection on the spread of COVID-19. To rely perhaps a bit too much on Greek myths, what began as a Herculean task now seems a Sisyphean one, given the vast numbers of papers and preprints on COVID-19. This update only goes through Feb 1, 2021, and thus misses the potential impact of immunity from infection and/or vaccination on the extent of symptomatic disease and transmission and similarly how these factors change by the dominant variant. I appreciate the authors reasoned approach to noting the massive heterogeneity in the studies reviewed and their caution in applying meta-analysis tools not appropriate for the massive variability in findings across studies. 

A primary takeaway seems to be that these types of reviews may be overwhelmed by both the tsunami of reports and the rapidly changing landscape with widespread infection and vaccination and shifting variants, where each of these factors may contribute to differences in results for each of the three questions this study focuses on. As such, my main comment for the authors is to clarify how much they expect variants and immunity from both vaccination and infection to play a role and whether their study is focused just on 'wild-type' or is generally a question that can be answered across the multiple dimensions of immune history, dominant variant, and potentially other elements. The abstract and paper end by mentioning 'wild-type' SARS-CoV-2. Given the D614G mutation that was dominant even within the timeframe of this review/metaanalysis, was this really only a study of wild-type SARS-CoV-2 (as the last lines of the abstract and paper suggest)? It seems that no prospective study could be done to look at these questions any longer for that strain, but what about for other strains? I'd be interested in the authors thoughts on what kinds of studies will be needed now, given widespread population immune experience from vaccination & infection, and repeated replacement of dominant variants. 

*** Reviewer #3: 

This manuscript provides an update to the previous systematic review and meta-analysis of asymptomatic COVID-19 infections. Using 94 studies, the authors found that 13-45% of infections are asymptomatic, although the authors note high between-study heterogeneity. While the paper is well written, in my opinion it does not add a lot to the growing body of literature that has been published since the previous version.

My major concern with this article is that the database was last updated a year back on February 2, 2021. To justify a publication as a living systematic review and meta-analysis, the database needs to be updated. Additionally, as the authors mention in the introduction, several recent meta-analyses on asymptomaticity have been published, providing more up to do date systematic reviews and meta-analyses of the literature. For example:

Ma et al. performed a similar analysis with their database updated on Feb 4, 2021. https://jamanetwork.com/journals/jamanetworkopen/fullarticle/2787098

Number of studies included in the meta-analysis of asymptomatic percentage: 95

Sah et al. Last database update: April 2021

https://www.pnas.org/content/118/34/e2109229118#sec-3

Number of studies included in the meta-analysis of asymptomatic percentage: 170

 I also have concerns about the analyses performed in the study which I highlight below:

Following my previous point, Table 1 indicates that only 13 studies published between January 2021 - June 2021 were included in the analysis which is probably because the database wasn't updated after February 2021. However, page 11 line 257-258 mentions "At the time of the latest search date,17 records were preprints, 14 of which had been published in peer-reviewed journals by 23 November 2021. Does this mean that another search was performed on 23 November 2021? If there was another search performed in November 2021, why weren't any studies published after June 2021 included in the analysis?

Definition of asymptomatic infection: The analyses include studies where COVID diagnosis was made based on antibody test rather than RT-PCR test. For example, the study Pirnay JP, 2020 observed four individuals with no symptoms, but only two of these were positive RT-qPCR. The other two were negative RT-qPCR but tested positive on antibody tests. The study itself notes "two individuals (M8 and M9), only the rapid antibody tests were positive, making it uncertain whether they actually had the disease. This manuscript assumes that these two individuals were asymptomatic. However, since the dates of exposure for these two individuals are uncertain, assuming that the positive antibody (but PCR negative) test is an indication of active infection is incorrect. As such, any study that used an antibody test to diagnose an asymptomatic infection should be excluded from the analysis.

Lack of symptom follow-up in studies included in the analysis: Some of the studies included in the analysis do not report an absence of symptoms at the end of follow-up, but despite this lack of data, these individuals are categorized as asymptomatic in this manuscript. For example, Bi Q, 2020 (reference 2, Table S2) mentions "At the time of the first clinical assessment …..and 17 (20%) of 87 had no symptoms." The study does not specify that these individuals remained symptom-free at the end of follow-up, and so cannot be categorized as asymptomatic. Such studies should be removed from the meta-analysis of asymptomaticity.

Bias arising due to over-representation of symptomatic cases: Since the studies included in the meta analysis were not designed to calculate asymptomatic percentage, there is an over representation of cases symptomatic cases in most studies, biasing the asymptomatic percentage towards a lower estimate. For example, Lee JY, 2020 investigated inpatients diagnosed with COVID-19 where 20% of the patients had severe disease. In such a setting, it is likely that the majority of patients were admitted because they were exhibiting symptoms and therefore asymptomatic individuals were likely under-represented. Additionally, surveillance studies would also suffer from this bias due to a higher willingness of symptomatic individuals to participate/provide consent. This issue has been highlighted and corrected by a recent meta-analysis (Sah et. al, 2021). Although performing a subgroup analysis by study design attempts to approach this issue, all study designs (except the ones where 100% of the population was sampled) will have this inherent selection bias. Without a correction, the estimates of asymptomaticity presented in this meta-analysis are unreliable. 

Additional analysis: Performing a subgroup analysis of asymptomatic proportion by publication date is likely to be biased due to the publication efficiency of different journals (it may take anytime between 1 month to 6m+ to publish an article). A more informative analysis would look at the last date of data collection mentioned in the study. 

*** Reviewer #4: 

This paper is clearly written and well organized. I suggest a minor revision. Meanwhile, some information shall be updated in this article.

[see attachment]

*** Reviewer #5: 

This is an interesting and useful systematic review and meta-analysis on the occurrence and transmission potential of asymptomatic and presymptomatic SARSCoV-2 infections. The methods including the search, study selection and data extraction, risk of bias assessment, synthesis of the evidence using meta and ggplot2 packages in R and random-effects models are mostly adequate. However, there are still a few major issues needing attention.

1) As the authors said in conclusion "this review does not provide a summary estimate of the proportion of asymptomatic SARS-CoV-2 across all study designs". With very wide interquartile ranges (IQR) in most of meta estimates which is largely due to extremely high heterogeneity between studies, it basically demonstrated that the current meta-analysis in this paper doesn't work or the way the evidence was synthesed may not be adequate or practical. With the overwhelming differences in attributes/factors in these 94 studies includes, it is extremely challenging and ambitious to do a meaningful meta analysis. Another substantial limitation in this study is that there is no information on Covid variants or vaccination status of paticipants, which may have a big impact of heterogeneity of the meta results and subsequently led to inconclusive results and findings. The sub group analyses showed some better and narrow IQRs which might indicate a direction of future work in meta-analysing these studies. One of useful features of this study is that authors identified quite a few difficulties, challenges and practical issues in conducting the meta-analysis on asymptomatic infections, however it would be better if the authors could go further (e.g., some relatively neat sub group meta analysis, or extra meta regression analysis) and to be constructive with more in depth and critical discussions to shed some lights in the tunnel.

2) Prediction interval. Prediction intervals were frequently used in the paper. However this measure is not standard in meta-analysis. I can see prediction intervals were calculated using the 3 previous meta-analyses results but not clear how they were calculated exactly. Basically, as this is a key element of this study, we need a detailed paragraph explaining what this prediction interval is for and what it means in practical terms and how it is relevant to the the summary estimate of meta-analysis. So far, these extremely wide prediction intervals are basically saying no to all the meta analysis results, but the quesion is, is it valid or appropriate doing so? 

3) To go deeper in finding out the factors contributing to heterogeneity and wide IQR in meta results, I would suggest that authors to carry out some extra meta regression analyses to identify which factors have impact on meta results, e.g., time, age, region, ethnicity, and different study designs and settings so that can give some solid suggestions for future research on this topic.

***

[LINK]

---

## [Decision Letter · Decision Letter 2]

8 Apr 2022

Dear Dr. Low,

Thank you very much for re-submitting your manuscript "Occurrence and transmission potential of asymptomatic and presymptomatic SARS-CoV-2 infections: update of a living systematic review and meta-analysis" (PMEDICINE-D-22-00260R2) for consideration at PLOS Medicine.

I have discussed the paper with our academic editor and it was also seen again by all reviewers. I am pleased to tell you that, provided the remaining editorial and production issues are fully dealt with, we expect to be able to accept the paper for publication in the journal.

[LINK]

Please let me know if you have any questions, and we look forward to receiving the revised manuscript.   

Sincerely,

Richard Turner, PhD

rturner@plos.org

Requests from Editors:

Please check the version numbers for consistency, e.g., "v4" at Github, "fifth update" at line 161 and "first three versions" in the acknowledgements. 

At line 83 (abstract), should that be "0.95"?

In table 1, should that be "sex" rather than "gender"?

At line 270, please make that "8 studies" and use this general style throughout, although numbers should be spelt out at the start of sentences. 

At line 396, please revisit "0.38 95% 0.16-0.64", which appears inconsistent with the numbers quoted in the abstract, for example. 

Please revisit reference 26, which may contain a superfluous close bracket.

Please remove the asterisks from reference 50.

Please make that "JAMA" in reference 71 and any other instances; and "PLoS ONE" in reference 134.

Please make that "US" in reference 92; and "UK" in reference 191.

Please check the journal name in reference 109.

Should "MFM" be removed from reference 153?

Comments from Reviewers:

*** Reviewer #1: 

Excellent review that I hope will be published soon.

I still have a concern about claiming this as a living review. It sets expectations from an end user perspective and as this will likely only be published a year after the end date of the last search.

But leave it to the authors if they think this is important.

*** Reviewer #2: 

I am satisfied with the authors' responses and believe the manuscript is acceptable for publication. 

*** Reviewer #3: 

The authors have satisfactorily addressed my previous concerns. To facilitate replication of results in the future, my only remaining suggestion is to include study characteristics described in Table 1 to individual studies listed in Table S2, including the study design, number of index cases excluded (if any), study region and age range of study participants. 

*** Reviewer #4: 

The authors have updated the review with 40 additional studies. All questions were answered in detail and necessary changes were made carefully.

I suggested this article be accepted.

*** Reviewer #5: 

Many thanks authors for their great effort to improve the manuscript. I am satisfied with the response and revision especially the meta regression analysis was nicely done. No further issues needing attention.

***

[LINK]

---

## [Editor Report · Decision Letter 3]

13 Apr 2022

Dear Dr Low, 

On behalf of my colleagues and the Academic Editor, Dr Ford, I am pleased to inform you that we have agreed to publish your manuscript "Occurrence and transmission potential of asymptomatic and presymptomatic SARS-CoV-2 infections: update of a living systematic review and meta-analysis" (PMEDICINE-D-22-00260R3) in PLOS Medicine.

PRESS

Sincerely, 

Richard Turner, PhD 

rturner@plos.org